# Asynchronous mixing of kidney progenitor cells potentiates nephrogenesis in organoids

Ashwani Kumar Gupta[1,2], Prasenjit Sarkar[3], Jason A. Wertheim[1,2,4,5,6,7], Xinchao Pan[8], Thomas J. Carroll[8] & Leif Oxburgh[9✉]

A fundamental challenge in emulating kidney tissue formation through directed differentiation of human pluripotent stem cells is that kidney development is iterative, and to reproduce the asynchronous mix of differentiation states found in the fetal kidney we combined cells differentiated at different times in the same organoid. Asynchronous mixing promoted nephrogenesis, and heterochronic organoids were well vascularized when engrafted under the kidney capsule. Micro-CT and injection of a circulating vascular marker demonstrated that engrafted kidney tissue was connected to the systemic circulation by 2 weeks after engraftment. Proximal tubule glucose uptake was confirmed, but despite these promising measures of graft function, overgrowth of stromal cells prevented long-term study. We propose that this is a technical feature of the engraftment procedure rather than a specific shortcoming of the directed differentiation because kidney organoids derived from primary cells and whole embryonic kidneys develop similar stromal overgrowth when engrafted under the kidney capsule.

[1] Comprehensive Transplant Center, Northwestern University Feinberg School of Medicine, Chicago, IL, USA. [2] Department of Surgery, Northwestern University Feinberg School of Medicine, Chicago, IL, USA. [3] Maine Medical Center Research Institute, Scarborough, ME, USA. [4] Department of Biomedical Engineering, Northwestern University, Evanston, IL, USA. [5] Simpson Querrey Institute for BioNanotechnology, Northwestern University, Chicago, IL, USA. [6] Chemistry of Life Processes Institute, Northwestern University, Evanston, IL, USA. [7] Department of Surgery, Jesse Brown VA Medical Center, Chicago, IL, USA. [8] Department of Molecular Biology, University of Texas Southwestern Medical Center, Dallas, TX, USA. [9] The Rogosin Institute, New York, NY, USA. ✉email: leo9022@nyp.org

Kidney injury is a common consequence of several prevalent conditions such as type 2 diabetes, cardiovascular disease, and systemic lupus erythematosus. In addition, drug toxicity, ischemia, and genetic mutations cause primary kidney injury. It is estimated that ~15% of adults have reduced kidney function (https://www.usrds.org), and these individuals are predisposed to end stage kidney disease, in which function is impaired to the point that basic physiological processes cannot be maintained. For these ~750,000 patients in the USA (https://www.usrds.org), dialysis is generally the first step, but the small molecule filtration and volume adjustment accomplished in routine dialysis does not compensate for all of the functions of the kidney tissue that have been lost. This is one important explanation for the discrepancy in survival between dialysis and kidney transplantation[1], although contributing factors are multifactorial. Currently, over 100,000 patients are waiting for a kidney transplant in the USA (https://optn.transplant.hrsa.gov) and based on statistics from 2018, it is anticipated that ~20% will receive an organ this year. Each day 13 people die waiting for a kidney transplant and there is a pressing need to seek new strategies to increase the availability of tissue.

Although we have known since the 1950s that mammalian kidneys can be cultured in vitro[2], attempts to grow transplantable tissue from primary cells and embryonic organs have met with limited success. Recent years have seen rapid advances in regenerative medicine following the discovery that adult somatic cells can be programmed back to induced pluripotent stem cells (iPSCs) from which their differentiation can be directed along any organ lineage. Theoretically, this provides two major advantages. First, it may be possible to recreate the complex interaction of cell types required for organ formation because multiple cell types can be differentiated simultaneously through directed differentiation. Second, tissues derived from iPSCs are autologous with the individual from which the iPSCs were reprogrammed, which should ensure minimal transplant rejection.

Procedures for the directed differentiation of human pluripotent stem cells (PSCs) to kidney cells have been developed[3–5], and in these procedures, PSCs are first differentiated to kidney progenitor cells, then triggered to epithelialize using a pulse of the GSK3 inhibitor CHIR, which is thought to mimic the natural epithelialization stimulus Wnt9b. This single epithelialization pulse differs from the process of differentiation in the developing kidney, where cells at numerous stages of differentiation coexist within the organ. Recent work has argued for a model in which the nephron forms through association of differentiating epithelial cells with cells that derive from the progenitor population,

suggesting a requirement for asynchronous cell populations for tissue formation[6].

We show that emulating this process in vitro through mixing of cells at distinct stages of differentiation promotes nephrogenesis, and we provide lineage-marking data proving that proximal and distal nephron components preferentially derive from different cell populations. When engrafted under the kidney capsule these heterochronic organoids are vascularized and display essential features of kidney tissue. However, overgrowth of stromal cells is an obstacle that prevents long-term study of these grafts, and we propose that this problem is a technical feature of the engraftment procedure rather than a specific shortcoming of the directed differentiation procedure as kidney organoids derived from primary cells and whole embryonic organs develop the same stromal overgrowth when engrafted under the kidney capsule.

## Results

**Strategy for kidney organoid tissue formation.** Directed differentiation of PSCs generates a mix of kidney progenitor cells that is predicted to represent the repertoire of cell types within the mesoderm that gives rise to the nephron, interstitium, and vasculature of the fetal kidney. We therefore reasoned that culturing these cells in the organotypic conditions developed for culture of the rodent kidney[7] would be a logical starting point for the formation of new human kidney tissue from PSCs. After performing the 9-day directed differentiation procedure for WTC11 iPSCs or H9 human embryonic stem cells (ESCs) as described by Morizane et al.[3,8], we dispersed the cells and aggregated them by resuspending the cells in a low volume to form a dense slurry and pipetting carefully into a droplet on the hydrophobic surface of a polycarbonate filter. At this point culture conditions were switched to those developed for primary nephron progenitor cells (NPCs)[9]; APEL 2 medium supplemented with protein-free hybridoma medium, BMP7, FGF9, and heparin. In preparatory experiments, we found that the addition of BMP7, FGF9, and heparin promote the viability of cultured cells generated by directed differentiation and we determined that these factors could be removed from the medium after the first 4 days of aggregate culture to reduce cost (Fig. 1a). The mix of cells resulting from directed differentiation is predicted to contain either no or only a very small number of progenitors for the collecting duct system[10], which is known to serve as the inducer of nephron epithelium differentiation[11]. In prior experiments we have provided the Wnt stimulus that induces mesenchyme-to-epithelium transition through treatment with the small molecule

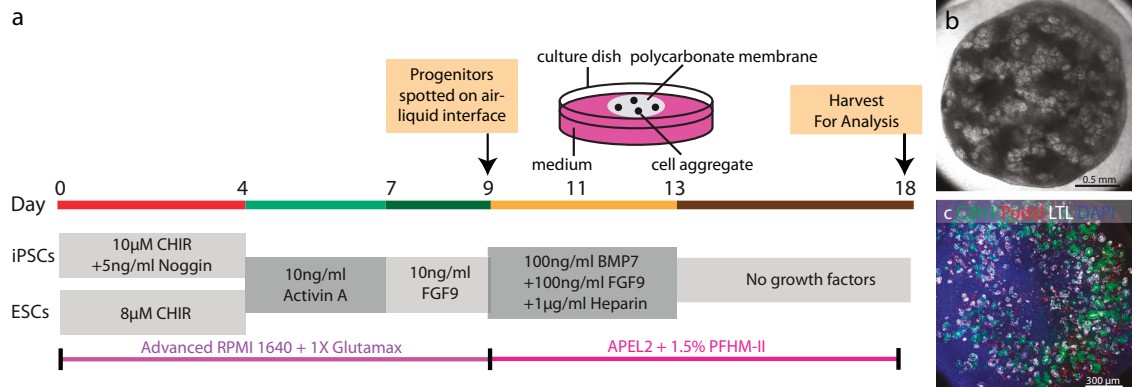

**Fig. 1 Generation of kidney organoids through directed differentiation. a** Schematic diagram showing directed differentiation of hPSCs into kidney progenitors through Morizane protocol and further aggregation to a kidney organoid. **b** Representative stereo microscope image of kidney organoid on day 18. **c** Immunofluorescence staining for Podocytes (Podxl), Proximal tubule (LTL), and distal tubule (Cdh1).

GSK3 inhibitor CHIR 99021[9,12]. However, in a series of titrations aimed at determining the dose of CHIR to use to initiate directed differentiation of WTC11 iPSCs to kidney progenitors, spontaneous epithelial differentiation was seen at day 9 of the directed differentiation, prior to detachment of cells and organoid formation (Supplementary Fig. 1a–g, l), and we therefore reasoned that cells may not require CHIR 99021 treatment for epithelial differentiation in aggregate culture. In a direct comparison between spontaneous differentiation and differentiation stimulated by a pulse of 3 μM CHIR for 48 h, we found that epithelial structures were more abundant and more tightly packed in spontaneously differentiating cell aggregates (Supplementary Fig. 1m, n). CHIR 99021 is a potent inhibitor of GSK3 and is expected to have multiple effects other than activating Wnt/β-catenin signaling. Thus, omitting it from our tissue differentiation protocol reduces the scope for unintended effects on the many different cellular differentiation pathways that must be activated, most of which are only partially understood at the level of cell signaling[13]. We found that kidney progenitor cells derived from directed differentiation of PSCs did indeed form epithelia following aggregation without CHIR treatment in our conditions (Fig. 1b, c). However, the epithelialization process does not appear efficient as large areas of undifferentiated cells can be seen in aggregates. For engraftment, we reasoned that it is important to convert the vast majority of cells in the aggregate to epithelium in order to avoid out-competition by undifferentiated cell types over the several-week period required for vascularization/perfusion, and we therefore developed the aggregate culture method further to enhance mesenchyme-to-epithelium transition.

Nephron formation in the kidney occurs in waves and undifferentiated NPCs are located immediately adjacent to epithelializing nephrons. Recent studies of human fetal kidney development suggest that there is gradual contribution of NPCs to the forming nephron, and that the differentiation fate of cells is dependent on when they are recruited into the nascent structure; time-dependent fate acquisition[6]. To establish a culture system in which NPCs could be added to epithelializing nephrons over time, we mixed newly differentiated cells with cells that had been aggregated on polycarbonate filters to allow formation of epithelia. To ascertain at which stages of differentiation it might be possible to improve epithelialization by this heterochronic recombination, we staggered directed differentiation protocols so that we could mix newly differentiated cells with cells that had been aggregated in organotypic conditions for 1, 2, or 3 days (Supplementary Fig. 2). We found that recombination of newly differentiated kidney progenitor cells with cells that had been aggregated and cultured for 2 days generated organoids with the most tightly packed epithelial structures (Fig. 2a–c). To quantify this effect, we compared the degree of staining for molecular markers of podocytes (podocalyxin), proximal tubule glycoproteins (lotus lectin), and distal tubules (cadherin 1) between organoids derived from an aggregate of a single population of cells from directed differentiation versus organoids from a heterochronic mix (Fig. 2d). Compared with aggregates cultured from only the first batch of directed differentiation cells or from only the second batch of directed differentiation cells, aggregates generated by heterochronic mixing displayed approximately double the number of structures stained for each molecular marker. On close examination, aggregates derived by heterochronic mixing contained proximal tubule cells (Fig. 2e), distal tubule cells (Fig. 2f), connecting segment or collecting duct (Fig. 2g), podocytes (Fig. 2h), endothelial cells (Fig. 2h), and stromal cells (Fig. 2i). Interestingly, the network of CD31-expressing presumptive endothelial cells in organoids from heterochronic mixing is more extensive and complex than that

seen in organoids generated from a single directed differentiation population (Supplementary Fig. 3).

In summary, our protocol efficiently generates organoids with tightly packed nephron epithelia and endothelial networks that are suitable for engraftment without the need for stimulation of organoid epithelialization by treatment with CHIR 99021 or other Wnt stimulators.

**Contributions of the two heterochronic cell batches**. To determine if the two batches of cells used for heterochronic recombination contribute equivalently to the different cell types in the epithelialized aggregate, we performed a series of experiments in which we combined H9 human embryonic stem cells with H9 cells that have been modified to express a fluorescently tagged H2B histone subunit under the control of the ubiquitous CAG enhancer-promoter. Fluorescently tagged H9 cells (H9-FP) were either incorporated as the first batch with unlabeled cells as the second batch (Fig. 3a) or as the second batch, with unlabeled batch 1 cells (Fig. 3b). H9-FP cells introduced in either batch 1 or batch 2 contribute to podocyte (Fig. 3c, d), proximal tubule epithelium (Fig. 3e, f), distal tubule epithelium (Fig. 3g, h), and interstitial cells (Fig. 3i, j). However, quantification reveals that batch 2 cells preferentially contribute to the podocyte (Fig. 3k) and proximal tubule (Fig. 3l), while batch 1 cells preferentially contribute to distal epithelium (Fig. 3m) and interstitial cells (Fig. 3n). This observation is consistent with microanatomical characterization of fetal kidneys showing a time-dependent fate acquisition of NPCs in which the podocyte population is added late in the formation of the nascent nephron[6]. To understand if the replenishment of NPCs by addition of a second heterochronic batch of cells might result in increased abundance of undifferentiated NPCs in differentiated organoids, we compared the frequency of cells positive for the NPC marker SIX2 and negative for the differentiation marker LHX1 in organoids derived from single batches of cells versus organoids derived from heterochronic mixes (Fig. 3o, p). We found that the proportion of undifferentiated SIX2+/LHX1− cells was lower in organoids from heterochronic mixes than in organoids from a single batch of cells (Fig. 3q), a finding that is consistent with our other observations showing that differentiation is most efficient in organoids from heterochronic mixes.

**Engrafted kidney organoids form vascularized kidney tissue**. To determine if kidney tissue derived from heterochronic recombination forms perfused tissue in vivo, we engrafted organoids under the kidney capsules of severely immunocompromised NSG mice. The scheme for organoid differentiation and engraftment is shown in Fig. 4a. Twenty organoids of 1–1.5 mm diameter and 0.25–0.5 mm thickness were engrafted into a single subcapsular site in each animal, and animals were sacrificed 3 weeks after engraftment. First, we performed a comparison of engraftment of organoids derived from a single directed differentiation batch with organoids derived from heterochronic differentiation. Quantification of structures stained with the proximal tubule marker LTL shows over two-fold more structures in heterochronic grafts versus single batch grafts (Supplementary Fig. 4), and further grafting experiments were conducted exclusively with heterochronic grafts. Extensive growth of engrafted tissue was seen (Fig. 4b–d), and kidneys were vibratome sectioned for whole mount molecular marker analysis. Staining with the endothelial marker CD31 revealed widespread vascularization of the graft, emanating from the host kidney (Fig. 4e). Higher magnification imaging reveals patent vessels connecting with clusters of WT1/Podxl-labeled presumptive podocytes (Fig. 4f). Glomeruli with characteristic morphology including complex

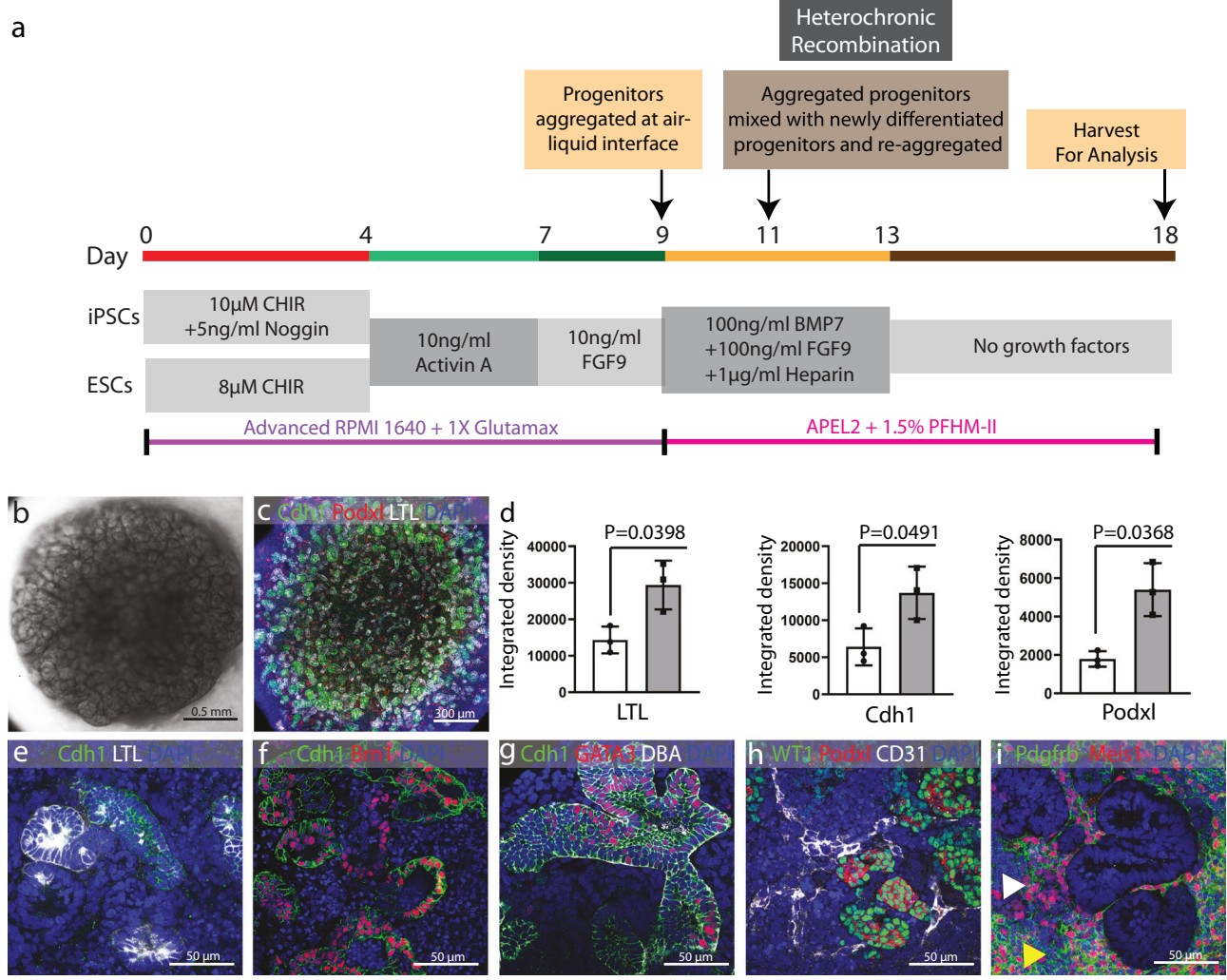

**Fig. 2 Generation of kidney organoids through heterochronic recombination. a** Schematic illustration of protocol for heterochronic mixing, in which two directed differentiations are staggered by 2 days so that the first batch may be aggregated and cultured for 2 days before the second, newly differentiated, batch is added. **b** Representative stereo microscope image of kidney organoid generated through heterochronic recombination. **c** Immunofluorescence staining for Podocytes (Podxl), Proximal tubule (LTL), and distal tubule (Cdh1) in the organoid. **d** Integrated density for LTL, Cdh1, and Podxl in organoids generated from single batch (open bar) and heterochronic mixing (gray bar). Values calculated from $n = 3$ independent biological replicates and expressed as mean ± s.d. Unpaired $t$ test with Welch's correction was applied to calculate $P$ value. **e–i** Representative high magnification immunofluorescence images of organoids derived from heterochronic recombination showing LTL+ Cdh1- proximal tubule, Brn1+ Cdh1+ distal tubule, CDH1+ GATA3+ DBA+ connecting tubule or collecting duct, Podxl+ WT1+ Podocytes, CD31+ endothelial network, Pdgfrβ+ Meis1- Pericytes (yellow arrow head), and Pdgfrβ+ Meis1+ stromal cells (white arrow head).

capillary structures and Bowman's space are found throughout the graft (Fig. 4g, h). Staining for the extracellular matrix proteins collagen I, collagen IV, and laminin reveals basement membrane morphology similar to glomeruli in the host (Supplementary Fig. 5); HNF4a and LTL label proximal tubule (Fig. 4i, j, k); cadherin1, Tamm Horsfall protein, and BRN1 mark distal tubule segments (Fig. 4j, k, l); GATA3, DBA, and KRT8 mark extreme distal tubule segments or collecting duct (Fig. 4m, n); Renin is expressed in subsets of cells at the base of glomeruli indicating differentiation of the juxtaglomerular apparatus (Fig. 4o). In summary, marker analysis showed that the cardinal cell types of the kidney are present in the graft, and that the tissue is vascularized. To ascertain which of the cells in grafts derive from human and which derive from mouse, we stained tissue for human nuclear antigen (Fig. 4p–u). Counterstain for the cell types listed above revealed that all cell types that differentiate in the subcapsular space derive from the human graft, with the exception of the CD31 and endomucin-expressing endothelium (Fig. 4t, u), which appears to be exclusively derived from the host.

**Tomography reveals perfusion 2 weeks after engraftment.** Having ascertained that organoids derived from heterochronic recombination of human kidney progenitors from directed differentiation are vascularized and maintained in vivo, we wanted to develop a method with which we could track the perfusion of grafts without having to sacrifice mice. This would enable longitudinal studies of graft retention and open up possibilities to monitor performance of the graft following drug treatment. Attempts to use MRI showed some promise but the long scan times required for imaging together with the high cost deterred us from pursuing this strategy. Instead, we used traditional X-ray technology with an iodinated contrast agent. To enable three-dimensional imaging, we developed settings for micro-computed tomography. Animals were imaged three times at 1-week

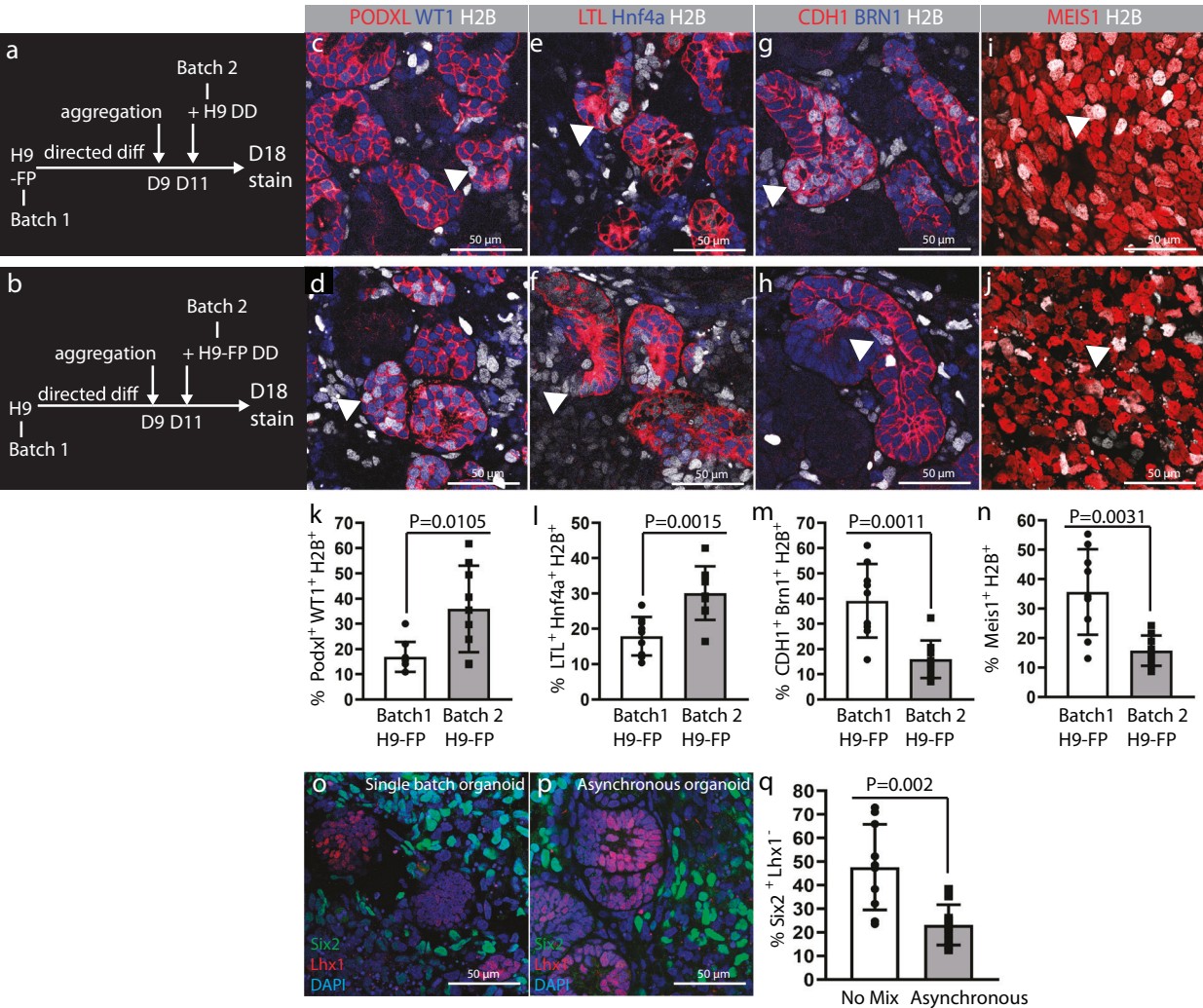

**Fig. 3 Contribution of heterochronic cell batches to nephron segments and interstitial cells. a** and **b** Experimental plans showing two strategies for heterochronic recombination of H9 and H9-FP cells, which constitutively express miRFP703 fused to histone H2B. **a** H9-FP cells were aggregated following directed differentiation and newly differentiated H9 cells were added 2 days thereafter. In **b**, H9 cells were aggregated following directed differentiation and H9-FP cells were added 2 days thereafter. **c** and **d** H2B-marked cell contribution to PODXL+/WT1+ podocytes. **e** and **f** H2B-marked contribution to LTL +/HNF4a+ proximal tubules. **g** and **h** H2B-marked contribution to CDH1+/BRN1+ distal tubules. **i** and **j** H2B-marked contribution to MEIS1+ interstitial cells. Arrowheads in each of the panels indicate H2B-marked cells that have contributed to the labeled cell populations. **k**–**n** shows quantification of contribution of H2B marked cells introduced in the batch 1 versus those introduced in batch 2 to each of the labeled cell populations. Cells in three random fields with differentiated structures were counted in each organoid. Values calculated from $n = 3$ independent biological replicates and expressed as mean ± s.d. Unpaired $t$ test with Welch's correction was applied to calculate $P$ value. **o**–**q** Immunostaining for SIX2, LHX1 on day 18 of differentiation comparing the frequency of SIX2+/LHX1− undifferentiated NPCs in single batch versus asynchronous mix organoids. **o** Example of single batch organoid on day 18. **p** Example of asynchronous organoid on day 18. **q** Quantification of cells in single batch versus asynchronous organoids. Cells in 10 random fields were counted in each organoid. Values calculated from $n = 3$ independent biological replicates and expressed as mean ± s.d. Unpaired $t$ test with Welch's correction was applied to calculate $P$ value.

intervals following engraftment using the procedure outlined in Fig. 5a. As expected, only calcified structures and kidneys are radio-opaque in animals following administration of contrast agent; note the absence of signal in the adrenal at the pole of the kidney (Fig. 5b). Although contrast agent circulates through soft tissues surrounding the kidneys, it does not accumulate there and the concentration is insufficient to generate a signal at the settings used. Accumulation of sufficient quantities of iodinated compound in the graft to generate signal would signify that the contrast agent is being cleared through the tissue and would provide both a confirmation that the tissue is perfused, and some measure of function. Interestingly, subcapsular grafts are not radio-opaque after 1 week (Fig. 5b), indicating that they are either not yet adequately perfused, or that they are not sufficiently

functional to accumulate iodinated compound. By 2 weeks after surgery (Fig. 5c), the graft has become diffusely radio-opaque confirming vascularization. By 3 weeks (Fig. 5d) the morphology of the contrast image has changed and there are regions of intense accumulation of iodinated compound within the graft, indicating functional compartmentalization within the graft.

To confirm connection of the graft vasculature to the systemic circulation, we administered a retro-orbital injection of FITC-labeled Griffonia Simplificata Isolectin B4 (FITC-IB4) to host animals immediately prior to sacrifice. FITC-IB4 binds glyco-proteins on the lumenal surface of endothelial cells, and can thus be used to confirm that systemic blood is circulating through graft vasculature. Whole mount imaging of vibratome sections of engrafted tissue revealed extensive networks of labeled vessels

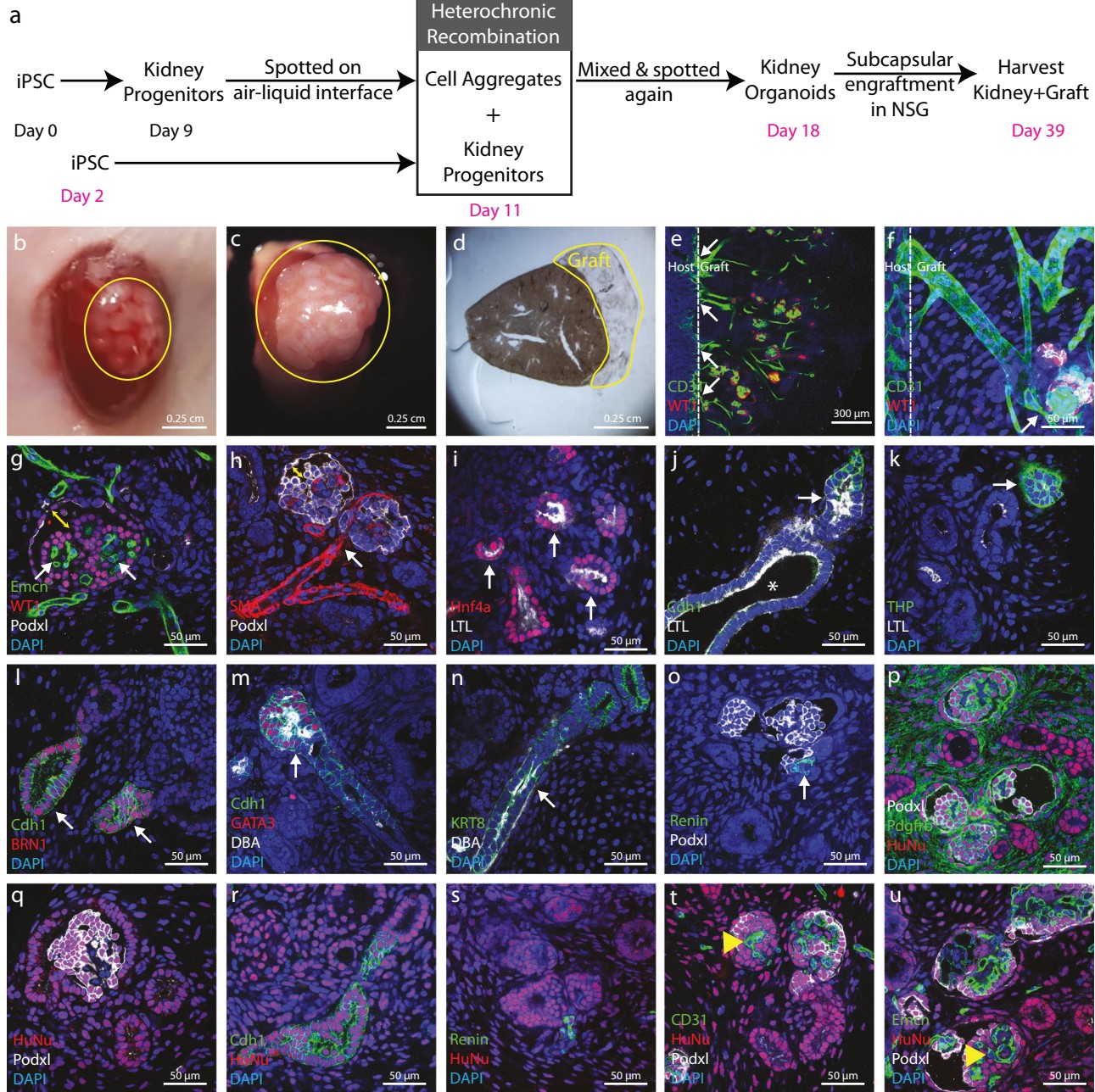

**Fig. 4 In vivo maturation of kidney organoids. a** Schematic diagrams showing generation of kidney organoids and their subcapsular engraftment in mice kidney. **b** Image showing kidney organoids immediately after engraftment. **c** Three weeks after engraftment organoids grows extensively in a big tissue. **d** Stereo microscope image showing vibratome section of mouse kidney with graft for further analysis. **e** Low magnification immunofluorescence image showing CD31+ endothelial network coming from host kidney widespread in the graft. **f** High magnification immunofluorescence image showing these CD31+ endothelial networks are connecting with Podxl+ WT1+ presumptive glomeruli. **g** and **h** Immunofluorescence images showing characteristic morphology of glomerular structure including Podxl+ WT1+ podocytes, Emcn+ capillary tuft and bowman's space (yellow double arrow head). These glomerular structures also shows direct connection with SMA+ vascular network. **i** Immunofluorescence image showing Hnf4a+ and LTL+ proximal tubule and **j** this LTL+ Cdh1− proximal tubule segmented in LTL− Cdh1+ distal tubule with lumen (white arrow head). **k** Immunofluorescence image showing THP+ loop of Henle and **l** Brn1+ Cdh1+ distal tubule. **m** and **n** Immunofluorescence image showing Cdh1+ GATA3+ DBA+ connecting tubule or collecting duct connected with Cdh1+ GATA3− DBA− distal segment of nephron. In addition, immunofluorescence images showing KRT+ DBA+ extreme distal segment or collecting duct. **o** Further, immunofluorescence image showing Renin+ juxtaglomerular cells at the base of Podxl+ glomerular structure. We did not observe Renin+ cells in organoid before engraftment which is another confirmation of organoid maturation in vivo. **p–s** Immunofluorescence image showing Pdgfrβ+ mesangial cells surrounded by Podxl+ Podocytes inside presumptive glomerular structures. Further immunofluorescence images showing differentiated Pdgfrβ+ mesangial or interstitial cells, Podxl+ podocytes, Cdh1+ epithelial structures and Renin+ cells were derived from human cells (HuNu+) but **t–u** Emcn+/CD31+ endothelial cells were derived from mouse (HuNu−, yellow arrow head). Images shown here are representative and were derived from *n* = 6 NSG mice.

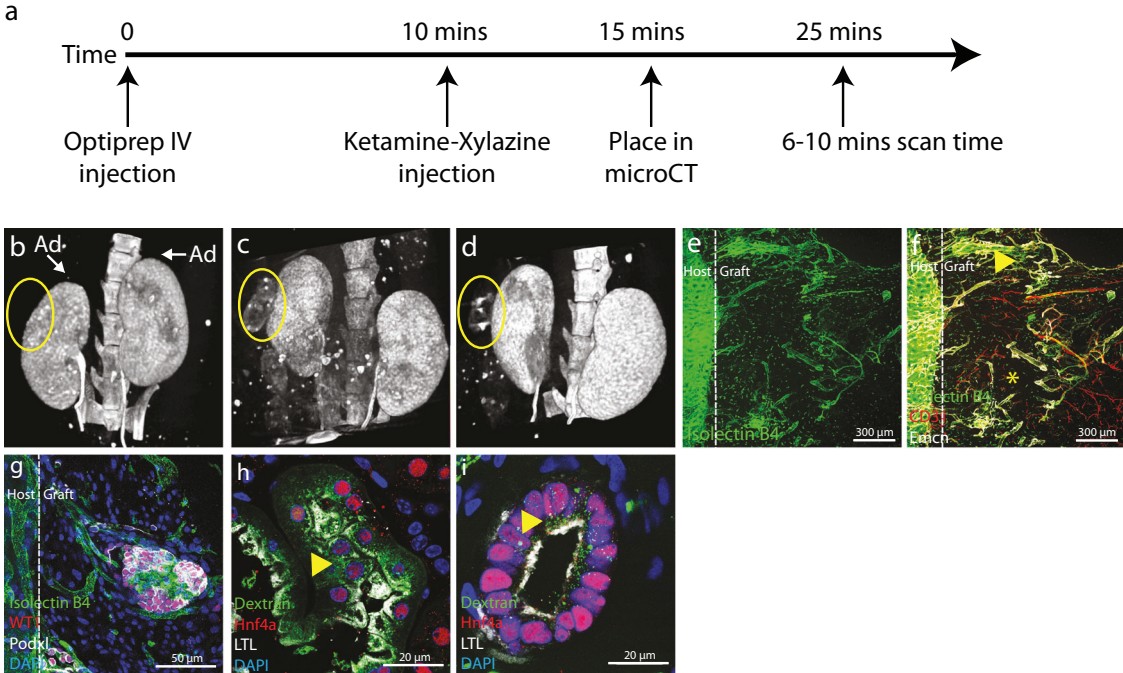

**Fig. 5 Micro-computed (μ-CT) tomography on live mice and functional characterization of the graft. a** Schematic strategy for micro-computed tomography. **b** μ-CT image 7 days after PSC organoid engraftment showing contrast agent accumulation only in the kidney of the host and not in the graft (circled). Note that there is no contrast agent accumulation in any abdominal soft tissues adjacent to the kidney, for example the adrenals (Ad). **c** μ-CT 14 days after engraftment shows diffuse accumulation of contrast agent in the subcapsular graft (circled) as well as in the host kidneys. **d** μ-CT 21 days after engraftment shows punctate contrast agent accumulation in the graft (circled) as well as accumulation in the host kidney. **e** Fluorescence image showing circulation of FITC-labeled isolectin B4 in a graft 3 weeks after subcapsular engraftment. FITC-labeled isolectin B4 was intravenously injected in the orbital sinus, and circulation into the graft implies that it is connected to the systemic circulation. **f** Co-staining of FITC-isolectin B4 perfused graft with the endothelial markers endomucin (Emcn) and CD31. Immunofluorescence image showing graft is compartmentalized into highly vascularized regions (yellow arrow head) and poorly vascularized regions (yellow asterisk), consistent with the μ-CT image 3 weeks after engraftment. **g** High power image of FITC-isolectin B4 costained with the podocyte markers WT1 and podocalyxin (Podxl) shows that presumptive glomeruli in the graft are connected to the systemic circulation 3 weeks after engraftment. **h** and **i** Fluorescently labeled dextran was injected into the orbital sinus 30 min before sacrifice to determine if proximal tubule epithelial cells of the graft take it up in vesicles. **h** Fluorescent dextran vesicle uptake (yellow arrowhead) in epithelial cells of the host that stain with the proximal tubule markers HNF4a and LTL. **i** Fluorescent dextran vesicle uptake (yellow arrowhead) in epithelial cells of the graft that stain with the proximal tubule markers HNF4a and LTL. Mice for this experiment were harvested 3 weeks after engraftment. All images shown in this figure are representative of 3 engrafted mice per experiment.

(Fig. 5e, f). CD31 antibodies used in this study are species specific (Supplementary Fig. 6), and only vessels stained with mouse-specific CD31 could be located within the graft tissue (Fig. 5f). Within vascularized regions of the graft, FITC-IB4-stained vasculature is intimately associated with WT1/PODXL-labeled clusters of presumptive podocytes (Fig. 5g), supporting systemic circulation through glomeruli in the graft. Systemic circulation through glomeruli and evidence of distended Bowman's capsules (Fig. 4g, h, p, q) indicates that blood is being filtered in the graft tissue. To test whether filtration might be occurring, we administered retroorbital injections of fluorescently labeled dextran to hosts and monitored uptake in the proximal tubules of the graft. Dextran (glucose) in the systemic circulation passes over the glomerular filtration barrier and is actively reclaimed by epithelial cells of the proximal tubule. An example of a proximal tubule from the host kidney is shown in Fig. 5h; note the accumulation of AF-488-dextran in vesicles. Similarly, HNF4a and LTL-labeled proximal tubule epithelial cells within the graft show AF-488-dextran uptake (Fig. 5i), although the size of vesicles appears more modest than that seen in the host kidney. Based on these findings, we conclude that there is circulation of systemic blood through the vasculature of the graft, that this blood is filtered, and that the proximal tubule is sufficiently

differentiated to actively reclaim glucose. Thus, the engrafted human kidney tissue derived from directed differentiation shows hallmark signs of function, providing an important proof of principle and motivating further development of this technology with the aim of functional testing in disease models. One issue that complicates the use of these grafts for functional testing is the overabundance of stromal cells, which are highly proliferative and in time may out-compete functional graft tissue.

**Primary cell engraftment reveals stromal overgrowth.** One important question in understanding the overabundance of stromal cells in grafts of PSC-derived kidney tissue is if this is a problem specific to tissue generated from PSCs, or if other aspects of the engraftment technique may promote stromal proliferation in implanted kidney tissue. To answer this question, we engrafted E15.5 embryonic kidneys subcapsularly in adult NOD SCID kidneys (Fig. 6a). Vigorous growth of the embryonic implant could be seen 1 month after engraftment (Fig. 6b, Supplementary Fig. 7), with abundant vascularized glomeruli (Fig. 6c, d, Supplementary Fig. 7). Immunostaining with the stromal marker PDGFRβ revealed widespread areas of stromal cells between glomeruli (Fig. 6e), suggesting that factors associated with subcapsular engraftment promote stromal abundance. To confirm

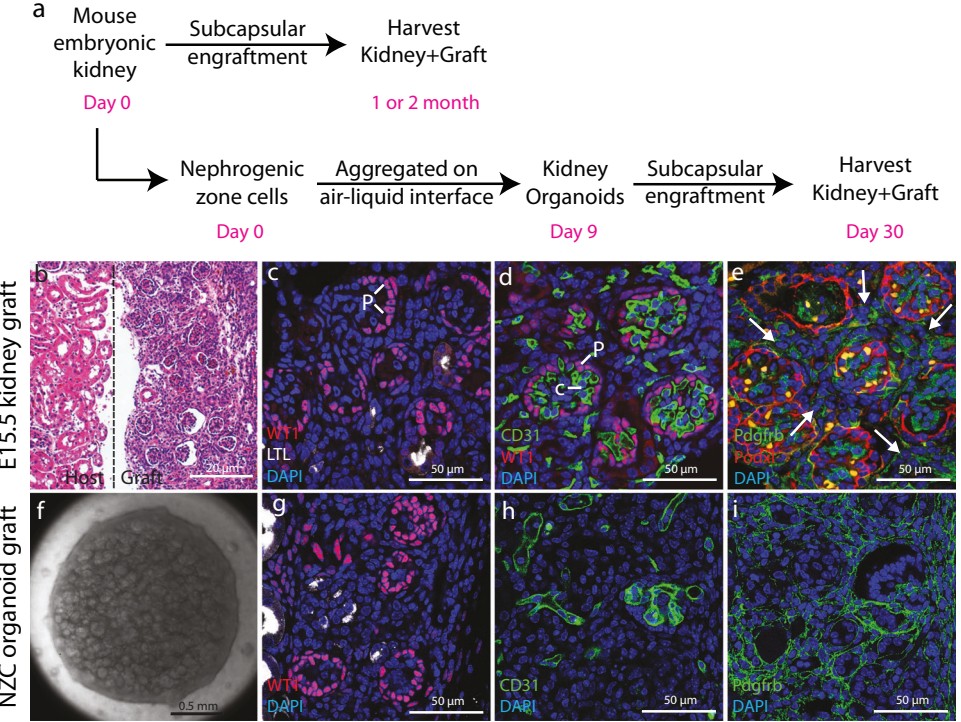

**Fig. 6 Subcapsular engraftment of embryonic kidney tissue results in stromal over-abundance. a** NSG mice were engrafted with embryonic kidney tissue using two distinct strategies: E15.5 whole embryonic kidneys were subcapsularly engrafted and harvested after 1 or 2 months, or nephrogenic zone cells were isolated from E17.5 kidneys, aggregated to organoids, subcapsularly engrafted and harvested after 2 weeks. **b–e** Tissue from E15.5 kidney graft. **b** H&E staining reveals extensive graft growth. **c** WT1 staining of podocytes shows abundant glomeruli and LTL staining for proximal tubules that these are sparse in the graft tissue. **d** CD31 staining of endothelial cells shows that glomeruli are vascularized. **e** Podocalyxin (Podxl) staining outlines glomeruli and PDGFRβ staining reveals abundant stromal cells between glomeruli. **f–h** Tissue from NZC organoid graft. **f** Example of NZC organoid at the time of engraftment. **g** WT1 staining for podocytes reveals abundant glomeruli. **h** CD31 staining for endothelial cells shows vascularization of the graft tissue. **i** PDGFRβ staining for stromal cells shows that these are abundant in the graft tissue.

this observation using a primary cell system more closely analogous to PSC-derived organoids, we used the nephrogenic zone cell isolation technique[14] to generate organoids composed of primary E17.5 nephrogenic zone cells using a similar aggregation and organotypic culture method as was used for PSC-derived directed differentiation cultures. Organoids derived from NZCs are morphologically very similar to those derived from PSCs (Fig. 6f). NZC organoids were subcapsularly engrafted in adult NSG mice and harvested after 3 weeks. At this point, extensive glomerulogenesis was noted (Fig. 6g), and the tissue showed abundant investment of CD31+ endothelial cells (Fig. 6h). However, staining for PDGFRβ again showed an over-abundance of stromal cells (Fig. 6i), demonstrating that the accumulation of stromal cells is not specific for kidney tissue generated by directed differentiation of PSCs.

## Discussion

This study shows that heterochronic mixing of cells derived by directed differentiation strongly potentiates the differentiation of kidney tissue in organoids derived from PSCs. NPC contribution to the nascent nephron of the developing kidney is asynchronous[6], and mixing kidney progenitor cells at distinct stages of differentiation is intended to emulate this process. The finding that the two different cell batches in a heterochronic mix contribute differentially to proximal and distal compartments of the nephron supports the idea that heterochronic mixing does indeed promote nephrogenesis by establishing a system for time-dependent differentiation. Interestingly, the addition of a second batch of cells does not result in the retention of residual undifferentiated NPCs in organoids derived from heterochronic

mixing. Rather, these cells appear to be more efficiently differentiated and organoids from heterochronic mixes contain fewer undifferentiated NPCs than organoids from single batches of cells at the time of engraftment.

Engrafted organoids differentiated using the heterochronic mixing strategy are efficiently vascularized and form characteristic nephron components in vivo. These are exclusively derived from the engrafted cells with the exception of the endothelium, which appears exclusively derived from the host. This feature of the engraftment is puzzling, since a well-developed network of endothelial cells is found in the organoid prior to engraftment. Since the complex glomerular vasculature can be derived from the host, this finding does not present a limitation, but rather raises the question why endothelial cells differentiated using our modification of the Morizane et al.[3] protocol do not contribute to blood vessels in the graft. Endothelial cells are known to be extremely fastidious in their growth requirements and protocols for differentiation of cells with potential to form functional vasculature in vivo differ from the conditions that we used. Another report has shown a mixed origin of endothelial cells in grafts of kidney organoids using the differentiation strategy developed by Takasato et al.[4,15]. Thus, it seems likely that our culture conditions do not provide key signals for differentiation of functional endothelial cells.

In vivo analysis of engrafted kidney tissue is essential for longitudinal studies, for example drug treatments. However, simple technology to detect and track subcapsular grafts without sacrificing the host animals has not been available. The use of micro-CT with iodinated contrast provides a rapid and relatively simple method for evaluating whether grafts have become

vascularized and if they are functionally concentrating contrast agent. In its current form, weekly scanning provides information on whether grafts have taken, their size and if they are displaying even enrichment of the contrast agent. In large studies comparing organoids generated from different patient iPSCs or iPSC with different genetic mutations, it will be important to interpret end-point analyses in the form of histology and functional measurements, such as AF-488-dextran incorporation in light of the dynamic changes in the graft over the engraftment period. There is a possibility that the administration of iodinated contrast agents has an effect on differentiation of grafts or on the host kidney that may affect the use of this technique for disease modeling. In our studies we did not notice a difference in histology between grafts that had been imaged and grafts that had not, and comparison of larger numbers of grafts will be required to detect subtle differences between these groups.

Stromal cells expressing appropriate molecular markers are found between epithelial structures in organoids prior to engraftment. However, this population expands excessively following engraftment. One possible interpretation would be that cells derived from directed differentiation are not stable in their differentiation state, leading to de-differentiation of epithelial cells to stroma. Genetic studies have for example shown that loss of expression of the transcription factor PAX2 causes NPCs to assume a stromal-like identity in the developing kidney[16]. Our finding that wild type embryonic kidneys engrafted under the kidney capsule show similar stromal accumulation suggests that this is not a feature of the directed differentiation process, but instead is associated with the micro-environment. One obvious difference between a subcapsular graft and a developing kidney is of course the collecting duct system. To date we have not observed any tubular connections between the graft and the host and based on this we suggest that degenerative changes in the graft are most likely due to the lack of urine outflow. Our future developments of engraftment technology will focus on establishing connections for urine outflow so that we can reach the goal of functional testing in animal disease models.

## Materials and methods

**Animals and regulatory compliance**. Male NSG mice (age = 6 weeks, from Jackson laboratory) were used for the engraftment and female NSG mice (age = 6 weeks, from Jackson laboratory) were used to generate NZCs organoids. Animal experiments were reviewed and approved by the Institutional Animal Care and Use Committees (IACUC) of Maine Medical Center and the University of Texas Southwestern Medical Center.

**Cell culture**. Human iPSC line WTC11 (a kind gift from Bruce Conklin, Gladstone Institute of Cardiovascular Disease) was maintained in mTeSR1 (Stemcell Technologies) on Matrigel (Corning)-coated six-well plates (Corning) in a 37 °C incubator with 5% $CO_2$. At 70–80% confluence, cells were dissociated with Accutase (Stemcell Technologies) and plated on Matrigel-coated six-well plates with mTeSR™1 containing 10 μM Rho kinase inhibitor, Y27632 (EMD Millipore). Y27632 was removed after the first 48 h and thereafter media was replaced with fresh mTeSR1 every day. Human H9 (purchased from Wicell) or H9-FP ubiquitously expressing miRFP703 fused to histone H2B (a kind gift from Dr. Andrew McMahon, University of Southern California) were maintained in StemFit (amsbio) with 100 ng/ml human FGF2 (R&D) on Geltrex (Thermo Fisher Scientific)-coated six-well plates. At 70–80% confluence, cells were dissociated with Accutase and plated on Geltrex-coated six-well plates with StemFit containing 100 ng/ml human FGF2 and 10 μM Y27632. After 48 h of culture, medium was replaced with fresh StemFit containing 50 ng/ml human FGF2. After the next 48 h, medium was replaced with fresh StemFit containing 25 ng/ml human FGF2.

**Directed differentiation and generation of kidney organoids**. We differentiated hPSCs to kidney progenitors according to Morizane et al.[3]. In brief, WTC11 were plated at a density of $1.4 \times 10^4$ cells/cm$^2$ in a six-well plate. On day 3, once cells became ~50% confluent, medium was replaced with Advanced RPMI 1640 (Thermo Fisher Scientific), 1× GlutaMAX™ (Thermo Fisher Scientific), 10 μM CHIR99021 (Reprocell), and 5 ng/ml Noggin (R&D Systems). On Day 4, medium was replaced with advanced RPMI 1640 containing 10 ng/ml Activin A (R&D) and 1X GlutaMAX™. On day 7, medium was replaced with advanced RPMI 1640

containing 10 ng/ml FGF9 (R&D) and 1X GlutaMAX™ for the next 2 days. H9 or H9-FP, cells were plated at a density of $1.7 \times 10^4$ cells/cm$^2$ in a six-well plate. On day 3, once cells became ~50% confluent, media was replaced with differentiation medium containing Advanced RPMI 1640, 1× GlutaMAX™ and 8 μM CHIR99021 (Reprocell). From day 4 to day 9, the same protocol was followed as for WTC11. On day 9 of directed differentiation, differentiated cells were designated "kidney progenitor cells". Kidney progenitor cells were harvested with TrypLE Express (Thermo Fisher Scientific) and resuspended at a density of $2.5 \times 10^5$ cells/μl in organoid initiation medium containing APEL2 (Stemcell Technologies), 1.5% PFHM-II (Thermo Fisher Scientific), 100 ng/ml FGF9, 100 ng/ml BMP7 (R&D Systems), and 1 μg/ml Heparin (Sigma-Aldrich). 1 ml/well of this medium was added to wells in a 24-well plate and isopore membranes (EMD Millipore) were suspended at the surface of the medium to create an air–liquid interface. Resuspended cells were spotted on top of the filter (2 μl/aggregate). Medium was changed every 48 h or when it turned yellow. All growth factors were removed from the medium 4 days after aggregation and organoids were cultured for a further 5 days with APEL2 containing 1.5% PFHM-II.

**Organoid generation through heterochronic recombination**. For heterochronic recombination of hPSCs, directed differentiation was performed on two batches of cells staggered 2 days apart. On day 9, the first batch of kidney progenitor cells was aggregated at the air–liquid interface as described above. Two days later, aggregated cells were gently broken into small cell clusters with a 200 μl micropipette and mixed with newly differentiated kidney progenitor cells. One dissociated aggregate was mixed with $5 \times 10^5$ kidney progenitor cells in 4 μl of organoid initiation medium and re-aggregated in two aggregates at the air–liquid interface. Medium was changed every 48 h or when it turned yellow. All growth factors were removed from the medium 4 days after aggregation and organoids were cultured for a further 5 days with APEL2 containing 1.5% PFHM-II.

**NZCs organoid generation and engraftment**. NZCs were derived from E17.5 mouse kidneys according to Blank et al.[14] and suspended at a cell density $2.5 \times 10^5$ cells/μl medium containing APEL2, 1.5% PFHM-II, 200 ng/ml of FGF9, and 1 μg/ml of Heparin. In a 24-well plate, 1 ml/well of medium was added per well and isopore membranes were suspended at the surface of the medium to create an air–liquid interface. NZCs were aggregated on top of the filter by spotting 2 μl per aggregate. Medium was changed every 48 h or when it became yellow. Five days after aggregation, all growth factors were removed and organoids were cultured with APEL2 containing 1.5% PFHM-II throughout for a further 4 days and then engrafted under the kidney capsule of adult mice. Mice were sacrificed 3 weeks after engraftment.

**Embryonic mouse kidneys for engraftment**. Pregnant dams were sacrificed at E15.5 and kidneys were dissected out of the embryos. Kidneys were then engrafted under the kidney capsules of adult NOD SCID mice (Charles River Laboratories). Graft recipients were sacrificed 1 month or 2 months after the engraftment and the kidneys were sectioned for H&E staining and immunostaining.

**Engraftment**. Organoids generated through heterochronic recombination were engrafted in NSG mice ($n = 6$ per experimental group). In brief, mice were anesthetized with isoflurane and an incision was made in the flank to exteriorize the kidney. A small incision was made in kidney capsule and organoids ($n = 20$) were placed under kidney capsule using fine blunt forceps.

**Micro-computed tomography**. Optiprep (Sigma) at a dose of 1.8 g/kg was injected retro-orbitally. Ten minutes after Optiprep injection, mice were anesthetized by intraperitoneal injection of a 16 mg/kg xylazine and 105 mg/kg cocktail. 25 min after optiprep injection, the torso of each mouse was scanned for 8–10 min at an isotropic voxel size of 76 microns (70 kV, 114 μA, 400 ms integration time) with a vivaCT 40 scanner (Scanco Medical Inc.). Two-dimensional gray-scale image slices were reconstructed into a three-dimensional tomography. Scans were reconstructed between the proximal end of L1 and the distal end of L5. The region of interest (ROI) for each animal was defined based on skeletal landmarks from the gray-scale images. Snapshots were taken at the best orientation to show graft perfusion.

**Perfusion of mice**. To study the connection between graft vasculature and the systemic circulation, 100 μl of 1 μg/μl FITC-labeled Griffonia Simplicata Isolectin B4 (Sigma) was injected into the retro orbital sinus. To evaluate glucose uptake of proximal tubule cells, 500 μl of 2 mg/ml Alexa Fluor 488-conjugated Dextran, 10,000 MW (Thermo Fisher Scientific) was injected into the retro-orbital sinus. 30 min after injection, mice were anesthetized with xylazine + ketamine and the right atrium of the heart was removed. Mice were then perfused via the left ventricle with 15 ml of 4% paraformaldehyde dissolved in PBS followed by 60 ml PBS. Kidneys were collected for analysis after perfusion.

**Immunostaining**. Whole mount staining was performed on organoids and 100 μm vibratome sections of kidneys carrying grafts. Organoids were fixed in 4%

**Table 1 List of antibodies/lectins.**

| Name | Catalog/clone number | Source | Dilution |
|---|---|---|---|
| Cdh1 | ab11512 | Abcam | 1:100 |
| Podxl | AF1658 | R&D systems | 1:100 |
| LTL | B-1325 | Vector | 1:200 |
| Brn1 | sc-6028-R | Santa cruz | 1:100 |
| GATA3 | 5852S | Cell signaling Tech. | 1:100 |
| DBA | B-1035 | Vector | 1:200 |
| WT1 | ab89901 | Abcam | 1:50 |
| Anti-human CD31 | 3528 | Cell signaling Tech. | 1:50 |
| Anti-mouse CD31 | 550274 | BD Pharmingen | 1:50 |
| Pdgfrβ | ab32570 | Abcam | 1:100 |
| Meis1 | 39795 | Active Motif | 1:100 |
| Hnf4a | C11F12 | Cell signaling | 1:100 |
| Lhx1 | 4F2-c | Developmental Studies Hybridoma Bank | 1:200 |
| Emcn | ab45771 | Abcam | 1:100 |
| SMA | C6198 | Sigma | 1:200 |
| THP | J65429 | Alfa Aesar | 1:200 |
| KRT8 | TROMA-1-c | Developmental Studies Hybridoma Bank | 1:200 |
| Renin | ab212197 | Abcam | 1:100 |
| HuNu | MAB1281B | Millipore | 1:50 |
| Collagen I | 600-401-103-0.1 | Rockland | 1:200 |
| Collagen IV | 600-401-106-0.1 | Rockland | 1:200 |
| Laminin | L9393 | Sigma | 1:200 |
| Six2 | 11562-1-AP | Proteintech | 1:100 |
| Donkey anti-Rabbit IgG, Alexa Fluor 488 | A-21206 | Thermo Fisher Scientific | 1:200 |
| Donkey anti-mouse IgG, Alexa Fluor 488 | A32766 | Thermo Fisher Scientific | 1:200 |
| Donkey anti-rat IgG, Alexa Fluor 488 | A-21208 | Thermo Fisher Scientific | 1:200 |
| Streptavidin, Alexa Fluor 488 | S11223 | Thermo Fisher Scientific | 1:200 |
| Donkey anti-Rabbit IgG, Alexa Fluor 568 | A10042 | Thermo Fisher Scientific | 1:200 |
| Donkey anti-Goat IgG, Alexa Fluor 568 | A-11057 | Thermo Fisher Scientific | 1:200 |
| Goat anti-Mouse IgG1, Alexa Fluor 568 | A-21124 | Thermo Fisher Scientific | 1:200 |
| Streptavidin, Alexa Fluor 568 | S11226 | Thermo Fisher Scientific | 1:200 |
| Donkey anti-Rabbit IgG, Alexa Fluor 647 | A-31573 | Thermo Fisher Scientific | 1:200 |
| Streptavidin, Alexa Fluor® 647 | S21374 | Thermo Fisher Scientific | 1:200 |

paraformaldehyde/PBS for 15 min at room temperature and then permeabilized with 1% Triton X100/PBS for 10 min at 4 °C. Blocking buffer consisting of 5% donkey or goat serum was added for 1 h and then primary antibodies diluted in blocking buffer were added (Table 1) and organoids were incubated at 4 °C overnight. They were then washed with PBS for 6 h at 4 °C and incubated with secondary antibodies (Table 1) and 0.01 μg/ml DAPI at 4 °C overnight. Organoids were then washed with PBS for 6 h and mounted in VECTASHIELD antifade mounting medium (Vector Laboratories). Fluorescent images were taken with a confocal microscope (Leica SP8). For quantification, integrated density of fluorescent images was measured using Image J. For each antibody stain, three separate fields from three different specimens were measured, and the mean was plotted with standard deviation (mean ± s.d.). Kidneys with engrafted organoids were pared down so that only a transverse slab of host kidney underlying the graft remained and this tissue was fixed with 4% paraformaldehyde/PBS for 1 h/mm thickness of the slab at 4 °C before vibratome sectioning at 100 μm. Immunostaining was performed using the same method as for organoids. For analysis of the embryonic kidney engraftment, kidneys were fixed overnight in 4% paraformaldehyde/PBS at 4 °C, paraffin embedded and sectioned. Sections were subjected to heat-induced epitope retrieval with Tris–EDTA, blocked in PBS with 0.1% Triton X-100 and 5% FBS and incubated overnight with primary antibodies. Secondary antibodies were added for 1 h at room temperature. Slides then were mounted with Vectashield and photographed by scanning laser confocal microscopy (Zeiss LSM-700). The primary antibodies used are shown in Table 1.

**Statistics and reproducibility**. Data in all graphs are expressed as mean ± s.d. and statistical analysis were done in Graphpad Prism version 8. P values were calculated by unpaired t test with Welch's correction for all graphs. In Supplementary Fig. 1l one-way ANOVA with Tukey's multiple-comparisons test was used. P values are given in all respective figures. Each dataset was derived from at least three independent biological replicates.

**Reporting summary**. Further information on research design is available in the Nature Research Reporting Summary linked to this article.

## Data availability

Data supporting the findings are available within the paper and its supplementary information. Source data file can be found in Supplementary Data 1.

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

## Acknowledgements

We would like to express our gratitude to Dr. Andrew McMahon and Tracy Tran at University of Southern California for sharing the H9-FP cells. The project described was supported by the National Institutes of Health grant number R24 DK106743 to L.O. and T.C. and Department of Veterans Affairs grant 01BX002660 to J.A.W. Core facilities support was provided by Maine Medical Center Research Institute core facilities for Molecular Phenotyping, Progenitor Cell Analysis, Microscopy, and Small Animal Imaging. A special thanks to Terry Henderson at the Maine Medical Center Research Institute Small Animal Imaging core for help with Micro-CT measurements. The content is solely the responsibility of the authors and does not necessarily represent the official views of the National Institutes of Health or Department of Veterans Affairs.

## Author contributions

A.K.G. contributed to experimental design, execution and interpretation of experiments, generation of figures and writing of the manuscript. P.S. contributed to experimental design. J.A.W. contributed to the execution of experiments. X.P. contributed to the execution of experiments. T.C. contributed to experimental design and interpretation. L.O. coordinated the project and contributed to experimental design, execution and interpretation of experiments, generation of figures, and writing of the manuscript.

## Competing interests

The authors declare no competing interests.
