## [Peer Review File · Communications Biology]

Editorial Note: *This manuscript has been previously reviewed at another Nature Research journal. This document only contains reviewer comments and rebuttal letters for versions considered at Communications Biology*

REVIEWERS' COMMENTS:

Reviewer #1 (Remarks to the Author):

The authors have adequately addressed the criticism by both reviewers. The manuscript has improved, however there is room for a little additional improvement. I have a few minor comments.

Minor comments:

- Include in picture 3O and P which picture represents single or asynchronous cell batches.
- Figure 3Q: is the quantification based on counting of single cells by immunofluorescence (e.g. nr of DAPI+ nuclei)?
- I am hesitant that by definition CHIR99021 causes off-target effects. There are few very specific inhibitors or activators, so would then all others cause off-target effects?
- What type of stromal cells do the authors observe in their transplanted tissues (e.g. also cartilage)?
- Figure 4T,U: the red is not so well visible and it seems many cells are negative
- Figure 4: dextran infusion. Can the authors provide an estimation of the number of tubular structures that contain FITC-dextran? Is it also visible in the glomerular structures?
- Figure 6: Can the authors make an estimation of the percentage of stromal tissue? Is the amount similar in all three tissues that were transplanted or was there a difference in the amount?
- Figure 6: an H&E staining of transplanted NZC organoids would be informative.
- Figure 6: There is no WT1 and pdx1 staining in Figure 6H and I. Can those be included or is there a specific reason why they are not included?

Reviewer #2 (Remarks to the Author):

The authors have improved their manuscript by carrying out some new experiments and providing further explanation and analysis of their results. Overall, this manuscript tables a modified kidney organoid differentiation protocol, by asynchronously mixing two different batches of NPCs, removing the CHIR pulse for epithelialization, and employing a temporal treatment of BMP7+FGF9+Heparin to boost NPC proliferation. The resultant kidney organoids present more abundant nephron-like structures (both in vitro and in vivo), and a more elaborated vascular network (in vitro). Furthermore, different batches of NPCs preferentially give rise to different nephron segments, emulating in vivo kidney development to certain degree. Upon implantation, these kidney organoids successfully anastomose with the host circulation system, further mature, and can perform some preliminary functions, such as glucose uptake. The authors also demonstrated that stroma overgrowth within implanted kidney organoids is not a phenomenon due to their iPSC origin. Instead, stroma overgrowth is generally seen in both implanted embryonic mouse kidney rudiments and in vitro cultured NPCs. Overall, the revised manuscript presents an alternative kidney organoid differentiation approach that

is likely to be useful. And the observation of stoma overgrowth within kidney organoid implants of multiple origins, though not explicitly explained at present, certainly warrants further investigation to resolve one of the major problems associated with kidney regeneration.

Reviewer #1 (Remarks to the Author):

The authors have adequately addressed the criticism by both reviewers. The manuscript has improved, however there is room for a little additional improvement. I have a few minor comments.

Minor comments:

- *Include in picture 3O and P which picture represents single or asynchronous cell batches.* These panels have been labeled.

- *Figure 3Q: is the quantification based on counting of single cells by immunofluorescence (e.g. nr of DAPI+ nuclei)?*

Yes, the figure legend has been revised to explain this.

- *I am hesitant that by definition CHIR99021 causes off-target effects. There are few very specific inhibitors or activators, so would then all others cause off-target effects?*

CHIR has become the reagent of choice in stem cell biology and directed differentiation because of its reliable activity in these systems. In many contexts, replacement of CHIR with LiCl or the small molecule BIO has been unsuccessful. It has not been explained whether this is a feature of differences in efficiency of interaction of each of these small molecules with their target proteins, or if this is a feature of the uptake of these small molecules by these complex cellular systems. Because CHIR is standard in the field, and because we have had very inconsistent outcomes using BIO and LiCl in stem cell systems (although they both work reliably in organ culture), we used CHIR in this study. The IC50 of CHIR for GSK3a is 10nM and for GSK3b is 6.7nM. These numbers are based on biochemical assays, and it is very interesting but unexplained why 500-1,000 fold higher doses than the IC50 are required for effects in cell culture. So, one potential cause of off-target effects is that we have an unknown but perhaps very high concentration of this inhibitor in treated cells causing effects on off-target kinases (and perhaps very different concentrations in the heterogeneous cell populations of directed differentiation if uptake/export differs between cell types). A bigger concern is that CHIR does not target any protein that is specific for the Wnt pathway; it inhibits Gsk3a and Gsk3b so can be considered a general GSK3 inhibitor. GSK3 is a central node in cytokine and insulin signaling, and regulates apoptosis (Bax), proliferation (Myc), and translation (eIF2B) (PMID: 24687186). We can assume that CHIR causes effects other than Wnt pathway activation based on the ubiquity of the fundamental cell biological pathways that it is regulating. Because CHIR is now the standard that all kidney directed differentiation protocols are based on we chose to use this reagent, to acknowledge its limitations and to exclude it where possible; interpreting its effect as Wnt-specific may obscure the understanding of which cell biological processes actually trigger the various steps of directed differentiation.

- *What type of stromal cells do the authors observe in their transplanted tissues (e.g. also cartilage)?*

Our characterization was limited to Pdgfr β + and/or Meis1+ stromal cells because we did not observe morphological evidence of cartilage. Anecdotally, we did see sporadic cartilage

formation in pilot studies of our engraftment technique; however, there were many variables in these initial surgeries and we did not include the pilot data because no reliable correlations can be made. All data included in this manuscript were based on a consistent surgical technique using cells differentiated from pluripotent stem cells that had been carefully characterized for stem cell pluripotency markers prior to use.

- *Figure 4T,U: the red is not so well visible and it seems many cells are negative*
The color has been corrected.

- *Figure 4: dextran infusion. Can the authors provide an estimation of the number of tubular structures that contain FITC-dextran? Is it also visible in the glomerular structures?*

Our main aim was to look FITC-dextran uptake in proximal tubules so we did not look for uptake in glomeruli. Because FITC-dextran uptake is sub-cellular we only imaged at high magnification, where only one or two tubules were visible in the field and we did not quantify what proportion of structures were positive so we can unfortunately not provide this data.

- *Figure 6: Can the authors make an estimation of the percentage of stromal tissue? Is the amount similar in all three tissues that were transplanted or was there a difference in the amount?*

Visually, all three tissue gave rise to similar amounts of stromal tissue, but we did not conduct a quantitative comparison so would prefer not to provide a speculative comparison in the text.

- *Figure 6: an H&E staining of transplanted NZC organoids would be informative.*

H&E staining was not conducted on transplanted NZC organoids – only on the engrafted embryonic kidneys. Unfortunately we cannot provide this data.

- *Figure 6: There is no WT1 and pdx1 staining in Figure 6H and I. Can those be included or is there a specific reason why they are not included?*

The anti-POXDL antibody that works in our hands is specific for human and does not stain mouse tissue, and we have therefore not used it for characterization of mouse grafts. Ant-WT1 and anti-PDGFR β are both rabbit monoclonals and cannot be combined. For this reason we decided to provide side-by-side images of the single marker stains for WT1, CD31 and PDGFR β in 6g-i. To generate a co-stain of WT1 and CD31 we would need to do lab work, which is not possible for the foreseeable future because all of our labs have been closed due to the ongoing coronavirus pandemic.